# LncRNA JPX Promotes Esophageal Squamous Cell Carcinoma Progression by Targeting miR-516b-5p/VEGFA Axis

**DOI:** 10.3390/cancers14112713

**Published:** 2022-05-31

**Authors:** Yi He, Rong Hua, Yang Yang, Bin Li, Xufeng Guo, Zhigang Li

**Affiliations:** Department of Thoracic Surgery, Section of Esophageal Surgery, Shanghai Chest Hospital, Shanghai Jiao Tong University, Shanghai 200030, China; hy2253@shchest.org (Y.H.); huarong@shchest.org (R.H.); yy2339@shchest.org (Y.Y.); libin@shchest.org (B.L.)

**Keywords:** esophageal squamous cell carcinoma, proliferation, angiogenesis, JPX, VEGFA

## Abstract

**Simple Summary:**

LncRNA JPX acts as an oncogenic regulator in various types of cancer. Here, we present insights into the mechanistic evidence for the function of JPX in ESCC progression. To clarify the potential role of JPX in ESCC, JPX was upregulated or downregulated in ESCC cells, and in a xenograft model. We showed that JPX promoted ESCC cell proliferation, migration, and invasion via the miR-516b-5p/VEGFA pathway. Our study revealed the importance of JPX as a promising biomarker for ESCC diagnosis and therapeutic target for ESCC in clinic.

**Abstract:**

Long non-coding RNAs (lncRNAs) are reported act as important regulators in various types of cancer. LncRNA JPX was identified as an oncogenic regulator in lung cancer. However, the function of JPX in the progression of esophageal squamous cell carcinoma (ESCC) remains unclear. In the present study, we found JPX was highly expressed in esophageal tissue from ESCC patients. Functional assays demonstrated that JPX promoted ESCC cell proliferation, migration, and invasion in vitro, and accelerated tumor growth in vivo. Mechanistically, the results showed that JPX functioned as a sponge of miR-516b-5p, which targeted vascular endothelial growth factor A (VEGFA) in ESCC cells. Interactions between miR-516b-5p and JPX or VEGFA were confirmed by luciferase reporter assays. Inhibition of JPX significantly attenuated the cell growth and mobility ability of ESCC cells in vitro. In addition, overexpression of miR-516b-5p abrogated JPX-enhanced proliferation, migration, invasion, and angiogenesis of ESCC cells. Our study demonstrated that JPX played an important role in promoting ESCC progression via the miR-516b-5p/VEGFA pathway, which might serve as a promising novel diagnostic biomarker and therapeutic target for ESCC in clinic.

## 1. Introduction

Esophageal cancer is one of the major causes of cancer-related death worldwide, which ranks seventh in terms of incidence and sixth in mortality worldwide [1]. Esophageal cancer is a common malignant tumor of the digestive tract. Approximately 50% of the cancer mortality rate derives from digestive tract cancers in China, including gastric cancer, liver cancer, and esophageal cancer [2]. Patients with esophageal cancer are mostly found and diagnosed due to dysphagia in the middle and late stages, and the 5-year survival rate is less than 20% after comprehensive treatment [3]. Esophageal cancer is mainly classified into esophageal adenocarcinoma (EAC) and esophageal squamous cell carcinoma (ESCC) [4]. Although the treatment of ESCC has been improved in recent years, the poor 5-year survival rate and prognosis are still disturbing to patients with ESCC owing to the early lymph node metastasis, the difficult eradication surgery, and the highly recurrence and metastasis rates [5,6,7]. Studying the molecular mechanism of ESCC progression is of great significance for improving ESCC treatment methods and strategies in clinic. 

Long non-coding RNAs (lncRNAs) are a class of non-coding RNAs with a length of more than 200 nucleotides [8]. LncRNAs have been proved to be closely related to the occurrence and development of tumors [9,10], involved in the regulation of chromatin remodeling, transcription, and post-transcriptional processes in the proliferation, invasion, and metastasis of cancer cells [11,12]. Previous studies have shown that lncRNAs functioned as the oncogenic regulator or suppressor of ESCC in vitro and in vivo [13,14]. A recent study has reported that lncRNA HERES promoted ESCC cell proliferation and metastasis by activating the Wnt signaling pathways via interaction with enhancer of zeste homolog 2 (EZH2) [15]. LncRNA CASC9 was revealed to be positively correlated with ESCC prognosis and metastasis, which upregulated laminin gamma 2 (LAMC2) expression by binding with CREB-binding protein (CBP) and modifying histone acetylation [16]. A novel lncRNA KLF3-AS1 was found to be downregulated in ESCC [17]. Upregulation of KLF3-AS1 inhibited ESCC cell invasion and migration by sponging miR-185-5p to inhibit KLF3 expression [17]. Latest studies have shown that a critical oncogenic lncRNA JPX promoted the development and progression of lung cancer [18,19]. JPX is a molecular switch for X-chromosome inactivation, which involves in the development of esophageal carcinoma [20]. The oncogenic role of JPX has also been reported in cervical, oral squamous cell carcinoma, and ovarian cancers [21,22,23]. However, the role of JPX in the progression of ESCC remains unclear. Here, the aim of this study is to investigate the function and molecular mechanisms of lncRNA JPX in ESCC. To clarify the potential role of JPX in ESCC, JPX was upregulated or downregulated in ESCC cells, and in a xenograft model. We showed that JPX promoted ESCC cell proliferation, migration, and invasion via the miR-516b-5p/VEGFA pathway, which might provide a promising novel diagnostic biomarker and therapeutic target for ESCC in clinic.

## 2. Materials and Methods

### 2.1. Patients and Tissue Samples

A total of 21 ESCC tissues and adjacent normal tissues specimens were collected from ESCC patients at Shanghai Chest Hospital (Shanghai, China). None of the patients with ESCC had received radiotherapy or chemotherapy before surgery. The clinical ESCC samples used in this study were histopathologically and clinically diagnosed in 2019. Written informed consent was obtained from all participants prior to sample collection. The clinical characteristics of patients were shown in Appendix A. TNM staging of ESCC patients was according to the American Joint Committee on Cancer (AJCC). This study protocol was reviewed and approved by the Ethical Committee and Institutional Review Board of the Shanghai Chest Hospital of the Shanghai Jiaotong University.

### 2.2. Cell Culture 

The human normal esophageal epithelial cell line Het-1A and human ESCC cell lines (KYSE150, KYSE450, Eca109, and EC9706) were obtained from the cell bank center of our institute. The cell line authentication was performed and the STR analysis of the cells was consistent with the STR data from China Infrastructure of Cell Line Resources database. Cells were cultured in RPMI-1640 (Gibco, New York, NY, USA) supplemented with 10% fetal bovine serum (FBS, Gibco, New York, NY, USA) and 1% penicillin/streptomycin (Invitrogen, Carlsbad, CA, USA) at 37 °C in a humidified atmosphere of 5% CO_2_.

### 2.3. Small Interfering RNA (siRNA) Synthesis, Plasmid Construction, and Transfection 

The miR-516b-5p mimics, si-JPX, si-VEGFA and the respective negative control (NC) were obtained from GenScript (Nanjing, China). pcDNA3.1 vector was obtained from YouBio (Changsha, China). JPX sequences were cloned into pcDNA3.1 vector. The cell transfection was performed using lipofectamine RNAiMAX (Invitrogen, Carlsbad, CA, USA) or Lipofectamine 3000 Reagent (Invitrogen, Carlsbad, CA, USA) according to the manufacturer’s instruction. In brief, cells were seeded in 6-well plate for 16 h. Amounts of 4 μg DNA, 100 pmol RNA, or 8 μL Lipofectamine 3000 was diluted by 250 μL serum-free medium. After incubation for 5 min, the samples were mixed gently and incubated for 20 min at room temperature. After different time points of transfection, cells were harvested for further detection.

### 2.4. Subcellular Fractionation

Cells were collected and used for nuclear and cytoplasmic protein extraction using nuclear and cytoplasmic extraction reagents (Life Technologies, Carlsbad, CA, USA). For nuclear and cytoplasmic RNA separation, cells were collected and extracted using PARIS™ kit (Life Technologies, Carlsbad, CA, USA).

### 2.5. Fluorescence In Situ Hybridization (FISH)

The FISH assay was performed as previously described [24]. The Cy3-labeled JPX probes used in our study were designed and synthesized by GenePharma (Shanghai, China). Briefly, the prepared cells were fixed with 4% paraformaldehyde containing 0.5% Triton X-100 for 20 min. The cells were incubated with probes at 37 °C overnight. The cell nuclei were stained with DAPI (Beyotime). The staining results were observed using Zeiss confocal laser microscopy (Zeiss 510, Jena, Germany). For tissue RNA FISH, fresh frozen sections were fixed in 4% paraformaldehyde, followed by incubation with RNase A at 37 °C for 45 min. Slides were prehybridized with probe overnight at 37 °C. After hybridization, sections were stained with DAPI (Beyotime). Slides were visualized using Zeiss confocal laser microscopy (Zeiss 510, Jena, Germany).

### 2.6. RNA Extraction and Quantitative Real-Time PCR (qRT-PCR)

Total RNA was isolated from tissues or cells using TRIzol reagent (Invitrogen, Carlsbad, CA, USA). The RNA was reversely transcribed to cDNA using a PrimeScript RT Master Mix Kit (Takara Bio, San Jose, CA, USA). A miScript SYBR Green PCR Kit (Qiagen, Hilden, Germany) was used to determine the gene expression using a 7900 real-time PCR system (Applied Biosystems, Life Technologies, Carlsbad, CA, USA). GAPDH or U6 was used as an internal control. Fold changes for the expression levels were calculated using the comparative cycle threshold (CT) method (2^−ΔΔCT^). The results were performed in triplicate in real-time quantitative RT-PCR. The primers used in the study were shown in Appendix A.

### 2.7. Western Blot

After transfection, cells were treated in a lysis buffer, followed by the extraction and quantification of proteins. Subsequently, the protein was transferred the into nitrocellulose membranes (NC membranes), blocked, and incubated in specific primary antibody at 4 °C overnight. The membranes were washed and incubated with second antibody for 2 h at room temperature. Finally, the protein bands were visualized using chemiluminescence detection system. The antibodies against VEGFA (ab51745, Abcam, Cambridge, UK), Fibronectin (ab2413, Abcam, Cambridge, UK), GAPDH (#14C10, Cell Signaling Technology, Danvers, MA, USA), N-cadherin (ab18203, Abcam, Cambridge, UK), Vimentin (ab3974, Abcam, Cambridge, UK), E-cadherin (sc-8426, Santa Cruz Biotechnology, Santa Cruz, CA, USA), and the secondary anti-IgG (ab150077, Abcam, Cambridge, UK) antibody were used to determine the protein expression levels. The intensity of protein was quantified using ImageJ software (NIH, Bethesda, MD, USA). For all uncropped original Western blot see Appendix A.

### 2.8. Cell Proliferation Assay

Cell proliferation was examined using a CCK-8 assay kit (Dojindo, Kumamoto, Japan). Briefly, cells were transfected with various vectors using Lipofectamine 3000 Reagent. After different time point of transfection, cells were added CCK-8 and incubated for 2 h. The absorbance at 450 nm was measured with a microplate reader (ELX800, BioTeK, Winooski, VT, USA).

### 2.9. 5-Ethynyl-2′-Deoxyuridine (EdU) Incorporation Assay

The EdU incorporation assay kit (RiboBio, Guangzhou, China) was used to evaluate cell proliferation. After transfection, cells were seeded into 96-well plates at a density of 1 × 10^4^ cells/well. 24 h after seeded, 50 μM EdU was added into each well and cells were incubated for 2 h at 37 °C, fixed with 4% paraformaldehyde, then stained with DAPI. The images were acquired using a fluorescence microscope (Olympus, Tokyo, Japan).

### 2.10. Transwell Migration and Invasion Assays

For the transwell migration assay, the above-transfected cells were plated to the upper chambers of non-matrigel-coated, 8-μm pore polyethylene membranes transwell plates (Corning, Tewksbury, MA, USA). For the matrigel-coated transwell invasion assay, pre-coated matrigel (BD Biosciences, San Jose, CA, USA) and transfected cells were placed in the upper chambers of transwell plates. cells were seeded on the upper chambers with serum-free medium. The lower chambers were added the medium contained 10% FBS as a chemoattractant. After culturing at 37 °C for 48 h, cells that appeared on the undersurface of the filter were fixed with methanol, stained with 0.1% crystal violet, and cells on the undersides of the filters were observed and counted in three randomly selected areas under a light microscope. 

### 2.11. Cell Cycle Analysis

Cells were seeded and transfected with si-JPX or si-NC for 48 h at 37 °C. The treated cells were washed by PBS and fixed in cold 70% ethanol for 30 min. After being washed by PBS, the cells were incubated with 400 µL PI for 30 min in the dark at room temperature. Finally, the DNA content was detected using a flow cytometer (BD Biosciences, San Jose, CA, USA) and the cell cycle was analyzed using FlowJo.7.6.1 software (FlowJo LLC, Ashland, OR, USA). 

### 2.12. RNA Pull-Down Assay 

The biotin (bio)-labeled JPX and control oligo probes (GenePharma, Shanghai, China) were incubated with magnetic beads (Life Technologies, Carlsbad, CA, USA) at 37 °C for 4 h. The RNA pull-down assay was carried out as previously described [25]. JPX overexpressing were lysed and incubated with the beads at 4 °C overnight. After washing with a wash buffer, the microRNA (miRNA) in the pull-down was determined using qRT-PCR.

### 2.13. RNA Immunoprecipitation (RIP) Assay

RNA-binding protein immunoprecipitation assay was performed by a Magna RIP Kit (Millipore, Temecula, CA, USA) according to the manufacturer’s methods. Cells were lysed by RIP lysis buffer, and magnetic beads with AGO2 (ab32381, Abcam, Cambridge, UK) or IgG (ab2410, Abcam, Cambridge, UK) antibody were prepared. The immunoprecipitation reactions were carried out by incubating the RIP lysate and the beads-antibody complex together with rotation overnight at 4 °C. Immunoprecipitated RNAs were purified and analyzed with qRT-PCR and normalized to the input control.

### 2.14. Luciferase Report Assay

The fragments contain the wild type (wt) and the mutant type (mut) of JPX or 3′UTR of VEGFA were respectively cloned into pmirGLO vector separately. EC9706 cells were co-transfected with miR-516b-5p mimics or mimics control and luciferase reporter. Twenty-four hours post transfection, cells were lysed using passive lysis buffer (promega, Madison, WI, USA) and the luciferase activity was measured using the dual-Luciferase reporter Assay System (Promega, Madison, WI, USA), and normalized to renilla luciferase activity, respectively. 

### 2.15. Tube Formation Assay

Seventy-five microliters of Matrigel (BD Biosciences, San Jose, CA, USA) was added into each well of a 48-well plate and allowed to solidify for more than 1 h at 37 °C. HUVECs were suspended in the indicated conditioned medium and seeded onto the gel. After 4 h of incubation, a light microscope was used to observe the tubular structures and acquire images. Tube formation was quantified by measuring the total length of the tubes using ImageJ (NIH, Bethesda, MD, USA).

### 2.16. In Vivo Experiments

Four-to-six-week-old male BALB/c nude mice were obtained from Slac Laboratory Animal (Shanghai, China) and maintained under specific pathogen-free conditions. Eight mice were divided randomly into two groups, and 1 × 10^6^ of EC9706 cells with or without JPX overexpression were injected into the tail vein of nude mice. The mice were sacrificed after 6 weeks, and the maximum and minimum length and weight of the tumors were measured. Tumor volume (mm^3^) was calculated with the formula: tumor volume (mm^3^) = longer diameter × shorter diameter^2^/2. The animal studies were approved by the Institutional Animal Care and Use Committee of Shanghai Chest Hospital, School of Medicine, Shanghai Jiaotong University.

### 2.17. Immunohistochemistry

Tumor sections were deparaffinized and rehydrated and the slides were incubated in hematoxylin for 2 min, washed with H_2_O, and then incubated in eosin for 2 min. The slides were imaged at 20× magnification using the Olympus microscope. The microvascular density was determined using anti-CD31 antibody (1:50, ab28364, Abcam, Cambridge, UK) in the paraffin-embedded tumor samples. In brief, paraffin sections were treated with xylene and hydrogen peroxide. The sections were incubated with primary and secondary antibody. The images were acquired by Olympus microscopy. The microvascular density of stained positively for CD31 was analyzed using ImageJ (NIH, Bethesda, MD, USA) and three fields were evaluated for each tissue (400×).

### 2.18. Statistical Analysis

The data were analyzed using Graphad Prism 5 software or SPSS 19.0 software (SPSS Inc., Chicago, IL, USA). The data were presented as the mean ± SD of at least three independent experiments. The statistical differences were calculated by the Student’s *t*-test or one-way ANOVA analysis of variance with Dunnett’s or Tukey’s test. Statistical significance was defined as *p* < 0.05.

## 3. Results

### 3.1. LncRNA JPX was Upregulated in ESCC 

To explore whether JPX exerted an impact on the progression of ESCC, we investigated the expression levels of JPX in ESCC tissues or adjacent normal tissues. The results showed that JPX levels were significantly highly expressed in ESCC tissues compared with those in adjacent normal tissues (Figure 1A). Moreover, the clinical analysis of JPX was determined in ESCC patients. The expression of JPX levels were significantly increased in patients with T_4_ or N_1_ stage of ESCC compared with those in T_2_/T_3_ or N_0_ stage of ESCC, respectively (Appendix A). We then investigated JPX expression in four esophageal cancer cell lines and a normal esophageal epithelial cell line. The results showed that the expression levels of JPX were significantly highly expressed in esophageal cancer cells, including EC9706, Eca109, and KYSE150, compared with those in normal esophageal epithelial cell line, Het-1A (Figure 1B). These results suggested that upregulation of JPX was involved in the progression of ESCC. To further investigate the role of JPX in ESCC, the subcellular distribution of JPX was detected in EC9706 cells. The results showed that JPX was mainly located in the nucleus of EC9706 cells (Figure 1C), which was further confirmed by FISH analysis (Figure 1D). In addition, the expression of JPX in ESCC tissues was upregulated in ESCC tissues compared with the normal tissues (Appendix A).

### 3.2. JPX Promoted Cell Proliferation and Tumor Growth in ESCC 

To investigate the biological function of JPX in esophageal cancer cells, JPX was knocked down by JPX siRNA (si-JPX). The efficiency of knockdown was confirmed in EC9706 and KYSE150 cells (Figure 2A). After downregulation of JPX for 48 h, the EC9706 and KYSE150 cell growth was significantly decreased compared with the control (Figure 2B,C). Next, we investigated the effects of JPX downregulation on cell proliferation and cell cycle, which might contribute to inhibit the ESCC cell growth. The results showed that the proliferative cells were significantly decreased after downregulation of JPX for 48 h in EC9706 (Figure 2D,E) and KYSE150 cells (Figure 2F,G). The cell cycle analysis demonstrated that downregulation of JPX led to cell cycle arrest in G1 phase and reduced the cell proportion in S phase in EC9706 (Figure 2H,I) and KYSE150 cells (Figure 2J,K), which consistent with the Edu proliferation results. These results suggested that downregulation of JPX might inhibit ESCC growth by reducing cell proliferating rate and arresting cell cycle. Subsequently, a xenograft model using EC9706 cells injection was used to validate the biological function of JPX in vivo (Figure 2L). Consistent with the results in vitro, overexpression of JPX significantly increased tumor volume (Figure 2M) and tumor weight (Figure 2O) compared with those in the control group. Moreover, HE staining results revealed that downregulation of JPX increased the ratio of nucleus to cytoplasm (Figure 2N). In addition, we found that the microvascular density was significantly increased in JPX-overexpressed EC9706 tumors (Figure 2P,Q). These results indicated that JPX might promote ESCC cell growth and proliferation in vitro and in vivo.

### 3.3. JPX Promoted ESCC Mobility In Vitro

To explore the role of JPX in the motility of esophageal cancer cells, the cell migration and invasion were determined after knockdown or overexpression of JPX. The results showed that downregulation of JPX significantly suppressed the migration and invasion of esophageal cancer cells (Figure 3A–D), whereas overexpression of JPX led to a significant increase in esophageal cancer cell migration and invasion (Figure 3E–H). Moreover, the effect of JPX on EMT-related markers was assessed following knockdown or overexpression of JPX. The results showed that downregulation of JPX increased the expression of epithelial marker E-cadherin and decreased the expression of mesenchymal markers, including Fibronectin, N-cadherin, and Vimentin (Figure 3I), whereas overexpression of JPX obviously decreased E-cadherin levels and increased Fibronectin, N-cadherin, and Vimentin levels (Figure 3J). In addition, the conditioned medium from esophageal cancer cells in HUVECs was used to evaluate angiogenesis activity in vitro [26]. The results showed that the relative length of tubes was significantly decreased following downregulation of JPX compared with the control, whereas overexpression of JPX led to a significantly elevated tube length compared with the control (Figure 3K,L).

### 3.4. JPX Functioned as a Sponge of miR-516b-5p in ESCC

To further investigate the mechanism of JPX-regulated esophageal cancer cell proliferation and mobility, the potential function of JPX as competing endogenous RNAs and act as a microRNA (miRNA) sponge was speculated for the distribution of JPX in the cell. Subsequently, the tentative miRNAs were predicted by the target prediction program (RegRNA, miRDB, and RNA22) analysis (Figure 4A). We identified ten overlapped miRNAs as potential direct downstream targets of JPX (Figure 4A). To determine the functional miRNA targets of JPX, a JPX-specific probe was constructed and used to JPX-overexpressed EC9706 cells. We then purified the JPX-associated RNAs by the specific probe and investigated the ten potential miRNA targets. As a result, miR-516b-5p was significantly enriched by the JPX-specific probe (Figure 4B), indicated that miR-516b-5p is a potential target of JPX in EC9706 cells. Moreover, the putative binding site of miR-516b-5p and JPX was shown in Figure 4C. To validate the binding potential, a luciferase reporter assay was performed and the results showed that overexpression of 10 nM miR-516b-5p significantly reduced the luciferase activity of the JPX wild-type vector, but failed to decrease that of the mutant vector (Figure 4D). The AGO2 immunoprecipitation assay showed that the AGO2 antibody was able to pull down both endogenous JPX and miR-516b-5p (Figure 4E,F), further validated their binding potential. Moreover, downregulation of JPX promoted miR-516b-5p expression in EC9706 and KYSE150 cells (Figure 4G). In addition, the miR-516b-5p levels were significantly lowly expressed in ESCC tissues compared with those in adjacent normal tissues (Figure 4H), indicated that miR-516b-5p may be negatively associated with JPX in ESCC. The clinical analysis showed that the expression of miR-516b-5p levels were significantly decreased in patients with T_4_ or N_1_ stage of ESCC compared with those in T_2_/T_3_ or N_0_ stage of ESCC, respectively (Appendix A).

### 3.5. MiR-516b-5p Abolished JPX-Promoted ESCC Growth and Mobility In Vitro

To examine the role of miR-516b-5p in esophageal cancer cell growth and mobility, miR-516b-5p was upregulated by transfection with miR-516b-5p mimics. The transfection efficiency of miR-516b-5p mimics was determined in EC9706 and KYSE150 cells (Figure 5A). Consistent with the effect of JPX knockdown on cell proliferation, overexpression of miR-516b-5p led to a decreased proliferation rate in EC9706 and KYSE150 cells (Figure 5B,C). Moreover, overexpression of miR-516b-5p reduced cell migration in EC9706 (Figure 5D,E) and KYSE150 cells (Figure 5F,G). Furthermore, cotransfection of JPX and miR-516b-5p abolished JPX-enhanced cell proliferation from 24 h to 96 h transfection in EC9706 cells (Figure 5H). Upregulation of miR-516b-5p inhibited the migration, and reversed JPX-increased cell migration in EC9706 cells (Figure 5I,J). Consistently, the tube length in HUVECs was suppressed by upregulation of miR-516b-5p. Cotransfection of JPX and miR-516b-5p abolished JPX-promoted cell tube length in HUVECs from the esophageal cancer cells conditioned medium in EC9706 cells (Figure 5K,L). These results indicating that JPX promoted esophageal cancer cell growth and mobility by sponging miR-516b-5p.

### 3.6. JPX Promoted Esophageal Cancer Cell Growth and Mobility via miR-516b-5p/VEGFA Signaling Pathway

To further explore the downstream signaling of miR-516b-5p involved esophageal cancer cell growth and mobility, the target genes were predicted by miRDB and TargetScan databases and we found that VEGFA was a potential target of miR-516b-5p. Upregulated expression of VEGFA was correlated with the progression of ESCC [27], indicated that VEGFA acted as an oncogene in ESCC. Consistently, the VEGFA levels were significantly highly expressed in ESCC tissues compared those in adjacent normal tissues (Figure 6H). The clinical analysis showed that the expression of VEGFA levels were significantly increased in patients with T_4_ or N_1_ stage of ESCC compared with those in T_2_/T_3_ or N_0_ stage of ESCC, respectively (Appendix A). Moreover, we found the positive coefficient between JPX and VEGFA in ESCC patients (Appendix A). The putative binding site of miR-516b-5p and 3′UTR of VEGFA was shown in Figure 6A. We found overexpression of 10 nM miR-516b-5p significantly reduced the luciferase activity of the wild-type 3′UTR of VEGFA vector, but failed to decrease that of the mutant 3′UTR of VEGFA vector (Figure 6B). Moreover, overexpression of miR-516b-5p or JPX knockdown significantly decreased VEGFA expression compared with the control in EC9706 and KYSE150 cells (Figure 6C,D). To validate the role of VEGFA in esophageal cancer cell function, VEGFA was downregulated using si-VEGFA transfection. The transfection efficiency of si-VEGFA was determined in EC9706 and KYSE150 cells (Figure 6E). As a result, downregulation of VEGFA led to a decreased proliferation rate in EC9706 and KYSE150 cells (Figure 6F,G). Furthermore, downregulation of VEGFA significantly decreased cell migration and invasion in EC9706 and KYSE150 cells (Figure 6I–N). In addition, we found JPX enhanced VEGFA protein expression, whereas cotransfection of JPX and miR-516b-5p suppressed JPX-promoted VEGFA protein expression in EC9706 cells (Figure 6O), indicated that JPX upregulated VEGFA expression via sponging miR-516b-5p. Collectively, these results suggest that JPX promoted esophageal cancer cell growth and mobility via miR-516b-5p/VEGFA signaling pathway. 

## 4. Discussion

ESCC is a malignant cancer with poor prognosis and 5-year survival rate owing to the eradication of unresectable, highly recurrence and metastasis rates [28]. Although tumor suppressors and oncogenes have been found to play a vital role in the occurrence and development of esophageal cancer [29,30], the fundamental molecular mechanism of ESCC carcinogenesis and development has not been fully elucidated. Therefore, understanding the molecular basis of ESCC progression and metastasis will provide effective therapeutic targets, screening, and early diagnosis biomarkers of ESCC. Here, we demonstrated that highly expressed JPX promoted the proliferation and mobility of ESCC cells via sponging miR-516b-5p to upregulate VEGFA expression. Moreover, overexpression of miR-516b-5p abrogated JPX-enhanced proliferation, migration, invasion, and angiogenesis in ESCC cells. Collectively, our results demonstrated that JPX promoted ESCC progression via the miR-516b-5p/VEGFA pathway, which might serve as a promising novel therapeutic strategy for the treatment of ESCC.

JPX is a nonprotein-coding lncRNA transcribed from a gene within the X-inactivation center with a length of 1696 bp, located on Xq13.2 [31]. A large number of studies have shown that JPX is abnormally expressed in a variety of malignancies, and it acts oncogenic regulator or suppressor relying on the cancer types [18,21,32]. However, the role of aberrant JPX expression in ESCC is still unknown. A previous study showed downregulation of JPX was associated with the poor prognosis and overall survival of hepatocellular carcinoma [32]. In contrast, our study found that a highly expressed JPX level in tissues of ESCC patients and overexpression of JPX promoted cell proliferation and mobility in ESCC cells in vitro and tumor growth in vivo, suggested that JPX exert critical function in the progression of ESCC by conferring more aggressive phenotypes on cancer cells. Consistently, upregulation of JPX has been demonstrated to expedite cancer cell proliferation, invasion, and metastasis in a variety of cancer types [18,22]. In accordance with the results in ESCC patients, we further found that the expression levels of JPX were significantly increased in various ESCC cell lines compared with those in normal esophageal epithelial cell.

To explore the effect of JPX knockdown or overexpression on ESCC cell function, we testified the JPX role in ESCC cell proliferation, migration, invasion, and angiogenesis. We found upregulation of JPX promoted ESCC cell proliferation, migration, and invasion in vitro and tumor growth in vivo. Downregulation of JPX led to cell cycle arrest in G1 phase, suggested that JPX may promote ESCC growth by facilitating cell cycle. Moreover, downregulation of JPX decreased EMT-related markers, including Fibronectin, N-cadherin, and Vimentin, whereas overexpression of JPX reversed their expression in ESCC cells, suggested that JPX promote EMT progression in ESCC cells. In line with these results, Pan J et al. demonstrated that JPX/miR-33a-5p/Twist1 axis activated EMT pathway, which correlated the invasion and metastasis of lung cancer [18]. In addition, overexpression of JPX stimulated angiogenesis activity of HUVECs in ESCC conditioned medium, indicated that JPX promote cell angiogenesis in vitro.

A number of studies have proposed that lncRNAs functioned as ceRNAs to affect tumor occurrence and development by regulating the corresponding gene expression through miRNA mediated pathway [33,34]. LncRNAs can sponge miRNAs by the same miRNA response elements (MREs), thereby regulating the target genes and ultimately affecting tumor progression [35]. Thus, we analyzed the tentative miRNAs by the RegRNA, miRDB, and RNA22 target prediction programs. Ten overlapped miRNAs were identified as the potential targets of JPX and miR-516b-5p was a direct target confirmed by RIP and luciferase reporter assays. On the other hand, miR-516b-5p was lowly expressed in ESCC patients, further demonstrated that miR-516b-5p might be negatively associated with JPX in ESCC. Interestingly, overexpression of miR-516b-5p suppressed cell proliferation, invasion, migration, and angiogenesis, whereas overexpression of miR-516b-5p and JPX abolished JPX-enhanced cell proliferation, migration, and angiogenesis in vitro. These findings revealed that downregulation of miR-516b-5p was also involved in the occurrence and progression of ESCC, suggested that miR-516b-5p can be used as an indicator for the diagnosis and therapeutic target of ESCC. In addition, VEGFA, a target of miR-516b-5p, was found to be coordinately highly expressed with JPX in ESCC cells. Previous studies have found that VEGFA was upregulated in various cancers, including non-small cell lung cancer, colorectal cancer, glioma, and breast cancer [36,37,38,39]. More importantly, upregulated expression of VEGFA was correlated with the progression of ESCC [27], indicated that VEGFA act as an oncogene in ESCC. In the present study, we demonstrated that VEGFA levels were highly expressed in ESCC tissues and downregulation of VEGFA could inhibit cell proliferation, migration, and invasion in ESCC cells. Moreover, JPX enhanced VEGFA protein expression, whereas cotransfection of JPX and miR-516b-5p suppressed JPX-increased VEGFA protein expression in ESCC cells, indicated that JPX upregulate VEGFA expression via sponging miR-516b-5p. These findings indicated that JPX play an oncogenic role via its interaction with miR-516b-5p/VEGFA pathway.

## 5. Conclusions

In summary, our results demonstrated that the JPX/miR-516b-5p/VEGFA axis may act as a new ceRNA regulatory network, participating in the angiogenesis and EMT pathway, thereby facilitating the progression of ESCC. These findings suggested that JPX and miR-516b-5p may serve as the potential therapeutic target and the novel biomarkers for the treatment and diagnosis of ESCC.

## Figures and Tables

**Figure 1 cancers-14-02713-f001:**
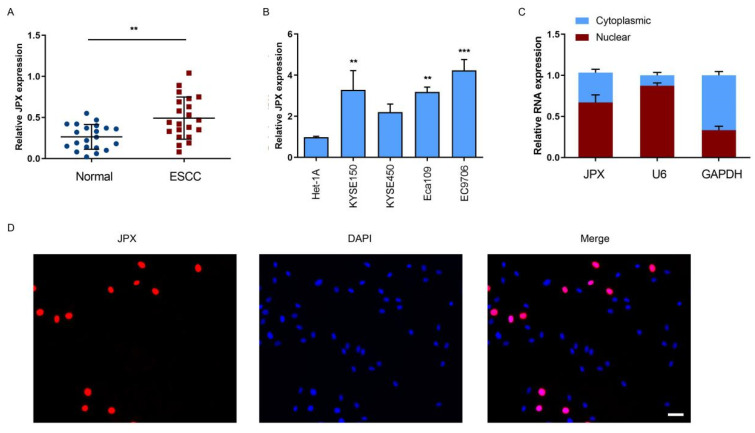
LncRNA JPX was upregulated in ESCC. (**A**) Expression levels of JPX in 21 ESCC tissues and adjacent normal tissues. (**B**) Expression of JPX in normal esophageal epithelial cell line Het-1A, ESCC cell lines (KYSE150, KYSE450, EC109, EC9706). (**C**) Expression level of JPX in the subcellular fractions of EC9706 cells. GAPDH and U6 were used as cytoplasmic and nuclear markers, respectively. (**D**) The location of JPX (red) in EC9706 cells was determined by FISH assay. Nuclei was stained by DAPI (blue). Scale bar, 50 μm. Data were mean ± SD. Statistical analyses were performed with Student’s *t*-test or one-way ANOVA followed by Dunnett’s test. ** *p* < 0.01, *** *p* < 0.001 versus Control.

**Figure 2 cancers-14-02713-f002:**
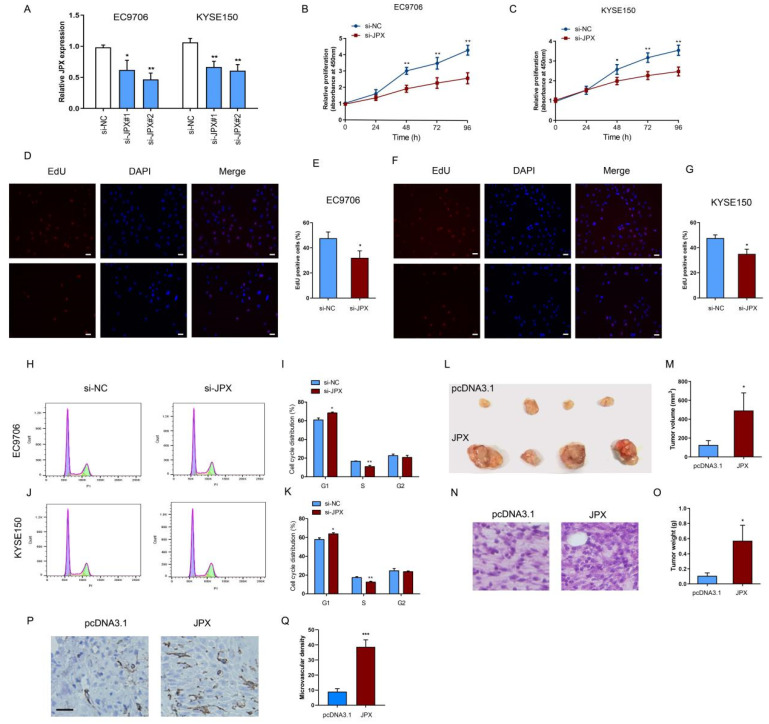
Downregulation of JPX inhibited esophageal cancer cell growth in vitro and in vivo. (**A**) Downregulation of JPX expression in EC9706 and KYSE150 using si-JPX. si-JPX#2 was used in the further study for its stable efficiency. Knockdown of JPX inhibited EC9706 (**B**) and KYSE150 (**C**) cell growth in vitro. EdU assays were used to detect the proliferation rate of EC9706 (**D**) and KYSE150 (**F**) cells after downregulation of JPX. Quantification of the EdU positive EC9706 (**E**) and KYSE150 (**G**) cells following downregulation of JPX. Columns are the average of three independent experiments. Scale bar, 25 μm. Cell cycle was determined in EC9706 (**H**,**I**) and KYSE150 (**J**,**K**) following downregulation of JPX by flow cytometry analysis. Columns are the percentage of cells in G1, S, and G2 phases, respectively. (**L**) Upregulation of JPX promoted esophageal tumor growth in vivo. Tumors developed from JPX transfected EC9706 cells showed significantly increased volume (**M**) and weights (**O**) compared with pcDNA3.1 control. (**N**) HE staining of esophageal tumor tissues from the different groups (200×). (**P**) CD31 immunohistochemistry of esophageal tumor tissues from the different groups. Scale bar, 50 μm. (**Q**) Quantification of microvascular density for CD31 staining in esophageal tumor tissues (n = 3). Data were mean ± SD. Statistical analyses were performed with Student’s *t*-test or one-way ANOVA followed by Tukey’s test. * *p* < 0.05, ** *p* < 0.01, *** *p* < 0.001 versus Control.

**Figure 3 cancers-14-02713-f003:**
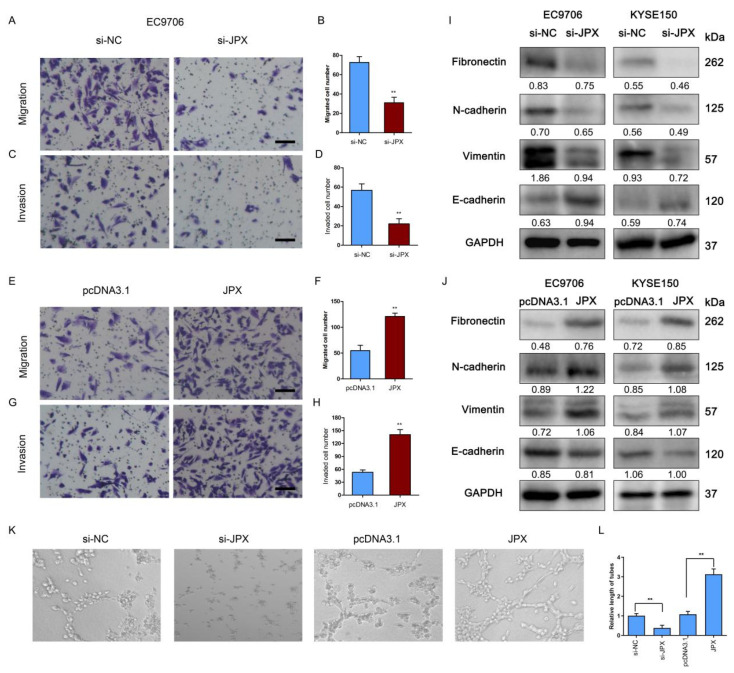
Downregulation of JPX inhibited esophageal cancer cell mobility in vitro. Downregulation of JPX inhibited EC9706 cell migration (**A**) and invasion (**C**). Scale bar, 25 μm. Quantification of the migrated (**B**) and invaded (**D**) cells following knockdown of JPX. Columns are the average of three independent experiments. Upregulation of JPX promoted EC9706 cell migration (**E**) and invasion (**G**). Scale bar, 25 μm. Quantification of the migrated (**F**) and invaded (**H**) cells following up-regulation of JPX. Columns are the average of three independent experiments. (**I**) Knockdown of JPX inhibited expression of EMT-related proteins in EC9706 and KYSE150 cells. (**J**) Upregulation of JPX enhanced expression of EMT-related proteins in EC9706 and KYSE150 cells. (**K**) Tube formation in HUVECs was inhibited by conditioned medium from EC9706 cells after JPX knockdown and was promoted by that from EC9706 cells after overexpression of JPX. (**L**) Quantification of the tubes length in HUVECs. Columns are the average of three independent experiments. Data were mean ± SD. Statistical analyses were performed with Student’s *t*-test or one-way ANOVA followed by Tukey’s test. ** *p* < 0.01 versus Control.

**Figure 4 cancers-14-02713-f004:**
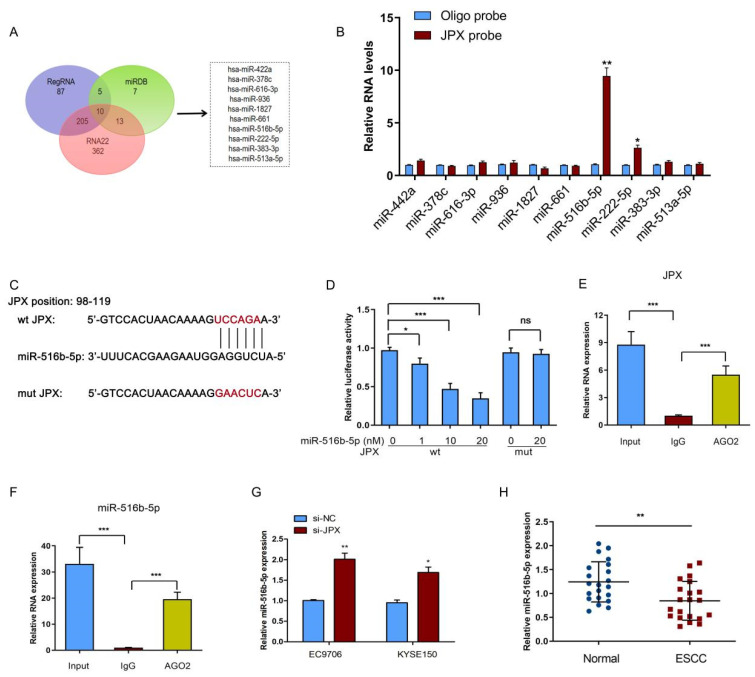
JPX functioned as a sponge of miR-516b-5p in ESCC. (**A**) Schematic illustration showing the overlapping interaction miRNAs of JPX predicted by RegRNA, miRDB, and RNA22 program analysis. (**B**) RIP assays were performed using JPX-overexpressed EC9706 cells with a JPX-specific probe and control oligo probe. qRT-PCR assays were performed to analyze potential miRNAs associated with JPX. The enrichment of JPX and potential miRNAs were normalized to that of the control probe. (**C**) Wild type (wt) and mutated type (mut) sequences of the putative binding sites between JPX and miR-516b-5p. (**D**) Dual-luciferase reporter assays were performed to validate the association of JPX and miR-516b-5p. The luciferase activity in EC9706 cells with or without miR-516b-5p overexpression and transfected with the wt or mut luciferase plasmids. (**E**,**F**) RIP and qRT-PCR assays were performed to validate the binding of JPX or miR-516b-5p to the AGO2 protein. (**G**) Expression levels of miR-516b-5p were determined in EC9706 and KYSE150 cells after downregulation of JPX. (**H**) Expression levels of miR-516b-5p were determined in 21 ESCC tissues and adjacent normal tissues. Data were mean ± SD. Statistical analyses were performed with Student’s *t*-test or one-way ANOVA followed by Tukey’s test. * *p* < 0.05, ** *p* < 0.01, *** *p* < 0.001.

**Figure 5 cancers-14-02713-f005:**
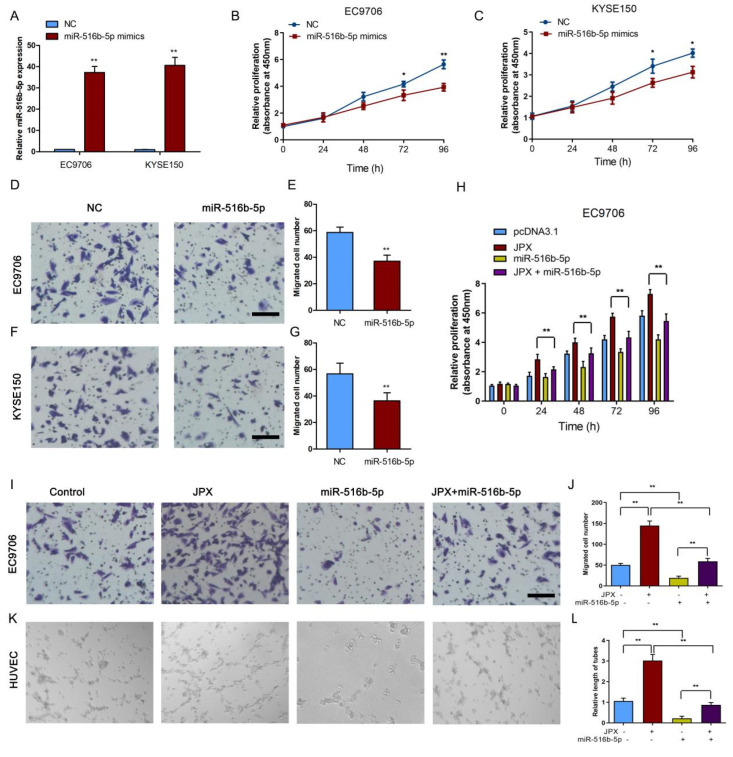
JPX promoted esophageal cancer cell growth and mobility by sponging miR-516b-5p. (**A**) Expression levels of miR-516b-5p were determined in EC9706 and KYSE150 cells after miR-516b-5p mimics treatment. Upregulation of miR-516b-5p inhibited EC9706 (**B**) and KYSE150 (**C**) cell proliferation in vitro. Upregulation of miR-516b-5p inhibited EC9706 (**D**) and KYSE150 (**F**) cell migration. Scale bar, 25 μm. Quantification of the migrated EC9706 (**E**) and KYSE150 (**G**) cells following up-regulation of miR-516b-5p. Columns are the average of three independent experiments. (**H**) Cotransfection of JPX and miR-516b-5p inhibited JPX promoted EC9706 cell proliferation. (**I**) The migration cells were determined following up-regulation of JPX and/or miR-516b-5p in EC9706 cells. Scale bar, 25 μm. (**J**) Quantification of the migrated cells following up-regulation of JPX and/or miR-516b-5p. (**K**) Cotransfection of JPX and miR-516b-5p reversed JPX-promoted EC9706 cell migration and tube formation of HUVECs. (**L**) Quantification of the tubes length in HUVECs. Columns are the average of three independent experiments. Data were mean ± SD. Statistical analyses were performed with Student’s *t*-test or one-way ANOVA followed by Tukey’s test. * *p* < 0.05, ** *p* < 0.01.

**Figure 6 cancers-14-02713-f006:**
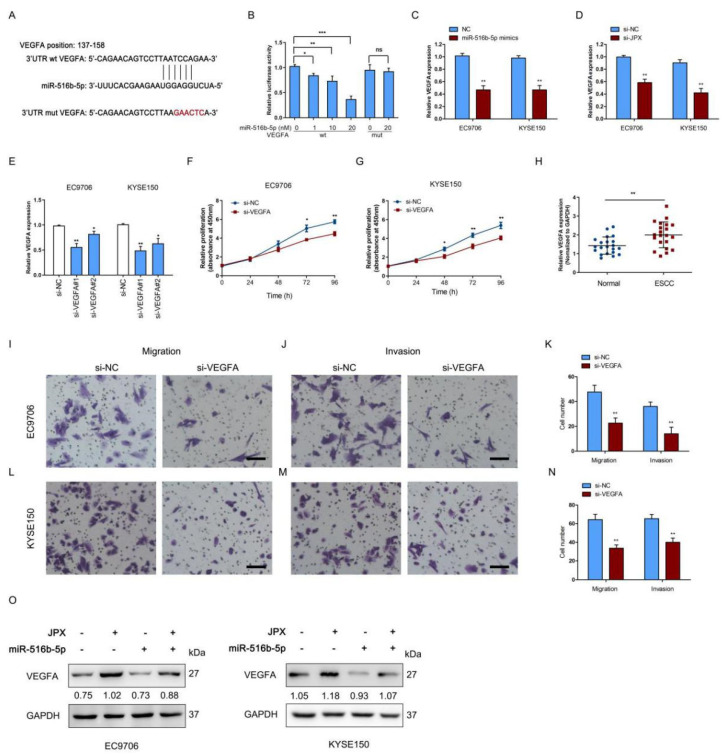
JPX promoted esophageal cancer cell progression by miR-516b-5p/VEGFA axis. (**A**) The predicted and the mutated binding sites of miR-516b-5p in the 3′UTR of VEGFA. (**B**) Dual-luciferase reporter assays were performed to validate the association of miR-516b-5p and VEGFA. The luciferase activity in EC9706 cells with or without miR-516b-5p overexpression and transfected with the VEGFA wt or mut luciferase plasmids. (**C**,**D**) Expression levels of VEGFA were determined in EC9706 and KYSE150 cells after overexpression of miR-516b-5p or downregulation of JPX. (**E**) Expression levels of VEGFA were determined in EC9706 and KYSE150 using si-VEGFA. si-VEGFA#1 was used in the further study. Downregulation of VEGFA inhibited EC9706 (**F**) and KYSE150 (**G**) cell proliferation in vitro. (**H**) Expression levels of miR-516b-5p were determined in 21 ESCC tissues and adjacent normal tissues. Downregulation of VEGFA inhibited EC9706 (**I**–**K**) and KYSE150 (**L**–**N**) cell migration and invasion. Columns are the average of three independent experiments. Scale bar, 25 μm. (**O**) Cotransfection of JPX and miR-516b-5p reversed JPX-enhanced VEGFA protein expression in EC9706 and KYSE150 cells. Data were mean ± SD. Statistical analyses were performed with Student’s *t*-test or one-way ANOVA followed by Tukey’s test. * *p* < 0.05, ** *p* < 0.01, *** *p* < 0.001 versus Control.

## Data Availability

The datasets used and/or analyzed in the study are available from the corresponding author on reasonable request.

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
