# Peer review of "LncRNA JPX Promotes Esophageal Squamous Cell Carcinoma Progression by Targeting miR-516b-5p/VEGFA Axis"

_cancers, 2022, doi:10.3390/cancers14112713_

Round 1
Reviewer 1 Report
As authors described, LncRNAs has been proved to be closely related to the occurrence and development of tumors. LncRNAs are involved in the regulation of chromatin remodeling, transcription and post transcriptional processing and other cellular biological processes playing a correspond role in the occurrence, development, proliferation, invasion and metastasis of cancer cells including ESCC cells. JPX is a nonprotein-coding lncRNA transcribed from a gene and abnormally expressed in a variety of malignancies, and it acts oncogenic regulator or suppressor relying on the cancer type. Thus, JPX and targeting miR-516b-5p is promising axis. However, the importance of this axis is well known in other types of cancer. The novelty seems to be limited. Based on the results about relationship between JPX/ miR-516b-5p and VEGFA, JPX upregulated VEGF expression via inhibition of miR-516b-5p. VEGFA is major angiogenic factor. There is no evaluation of angiogenesis though authors showed JPX overexpression tumor (Figure 2L). Moreover, there is not enough information about upstream of JPX. It is unclear when or how the expression level of JPX is changed, which makes readers difficult to understand of big picture. The usefulness of controlling JPX/ miR-516b-5p/ VEGFA axis is also unclear because anti-VEGF therapy is not established in ESCC. These points are concerns for publication.
Author Response
As authors described, LncRNAs has been proved to be closely related to the occurrence and development of tumors. LncRNAs are involved in the regulation of chromatin remodeling, transcription and post transcriptional processing and other cellular biological processes playing a correspond role in the occurrence, development, proliferation, invasion and metastasis of cancer cells including ESCC cells. JPX is a nonprotein-coding lncRNA transcribed from a gene and abnormally expressed in a variety of malignancies, and it acts oncogenic regulator or suppressor relying on the cancer type. Thus, JPX and targeting miR-516b-5p is promising axis. However, the importance of this axis is well known in other types of cancer. The novelty seems to be limited. Based on the results about relationship between JPX/ miR-516b-5p and VEGFA, JPX upregulated VEGF expression via inhibition of miR-516b-5p. VEGFA is major angiogenic factor.
There is no evaluation of angiogenesis though authors showed JPX overexpression tumor (Figure 2L).
Answer: We have added the microvascular density results in JPX-overexpressed EC9706 tumors in figure 2P-Q.
Moreover, there is not enough information about upstream of JPX. It is unclear when or how the expression level of JPX is changed, which makes readers difficult to understand of big picture.
Answer: JPX is a molecular switch for X-chromosome inactivation, which involves in the development of esophageal carcinoma (20). We have added the description in the introduction (Line 91-93). The upstream mechanism of JPX need to further explore.
The usefulness of controlling JPX/ miR-516b-5p/ VEGFA axis is also unclear because anti-VEGF therapy is not established in ESCC. These points are concerns for publication.
Answer: Angiogenesis during tumorigenesis contributes to the aggressiveness and poor prognosis of ESCC. Neovascularization supplies oxygen and nutrients to proliferative tumor cells, and serves as a conduit for migration. Understanding molecular mechanisms that contribute to angiogenesis may help identify the biological basis of ESCC and improve therapy. Targeting oncogenes involved in angiogenesis is needed to treat advanced ESCC. A number of studies have shown angiogenesis was a critical target in ESCC tumorigenesis and metastasis (1-4). Thus, we focused on the VEGFA as the key downstream factors in this study.
- Yang QS, Zhao JF, Qin YR, Xu LY, Li EM, Liao HX, Li B, He QY.Direct Targeting of CREB1 with Imperatorin Inhibits TGFβ2-ERK Signaling to Suppress Esophageal Cancer Metastasis. Adv Sci. 2020;7(16):2000925.
- Chen Y, Wang D, Peng H, Chen X, Han X, Yu J, Wang W, Liang L, Liu Z, Zheng Y, Hu J, Yang L, Li J, Zhou H, Cui X, Li F.Epigenetically upregulated oncoprotein PLCE1 drives esophageal carcinoma angiogenesis and proliferation via activating the PI-PLCε-NF-κB signaling pathway and VEGF-C/ Bcl-2 expression. Mol Cancer. 2019;18(1):1.
- Wang Y, Lu J, Chen L, Bian H, Hu J, Li D, Xia C, Xu H.Tumor-Derived EV-Encapsulated miR-181b-5p Induces Angiogenesis to Foster Tumorigenesis and Metastasis of ESCC. Mol Ther Nucleic Acids. 2020;20:421-437.
- Takahashi K, Asano N, Imatani A, Kondo Y, Saito M, Takeuchi A, Jin X, Saito M, Hatta W, Asanuma K, Uno K, Koike T, Masamune A.Sox2 induces tumorigenesis and angiogenesis of early-stage esophageal squamous cell carcinoma through secretion of Suprabasin. Carcinogenesis. 2020;41(11):1543-1552.
Reviewer 2 Report
The Authors tested the role and function of long non-coding RNA JPX (lncRNA JPX) in the biology of esophageal squamous cell carcinoma (ESCC). The Authors showed that this oncogenic regulator is upregulated in EC9706, Eca109, and KYSE150 compared to normal esophageal epithelial cell line, Het-1A ESCC. The JPX expression is evident in the cytoplasm. The functional assays included in the experimental design showed that JPX knockdown decreases ESCC:
(1) proliferation reducing the number of activly dividing cells in the S phase of the cell cycle and causing their shift toward G0/G1 phase (assays performed on EC9706 and KYSE150);
(2) mobility and invasion what was associated with lowered protein levels of fibronectin, N-cadherin, and vimentin simultaneously with increased levels of E-cadherin
In turn, the upregulation of JPX promoted esophageal tumor growth in vivo and tumor migratory potential.
Furthermore, Authors showed that the potential target for lncRNA JPX is miR-516b-5p. This small non-coding RNA inhibits JPX promoted proliferation and reversed JPX increasing cell migration. Finally, the Authors explained the mechanism of JPX action, showing that this lncRNA promotes esophageal cancer cell growth and mobility via miR-516b-5p/VEGFA signalling pathway.
The article is well-performed, with an excellent experimental design that gives a proof of concept. Indtoduction provides insight into the topic. Methods are adequately described.
The data description and presentation are good. However, the presentation of the data must be improved:
1) Please provide original blots with adjusted molecular weight markers
2) Please provide information regarding the molecular weight of detected proteins
3) Figure 1D: the quality of microphotographs is poor, especially EdU staining; the scale bar is bearly seen.
4) Figure 3, 5 and 6: please provide the scale bar for the microscopic images.
Author Response
The Authors tested the role and function of long non-coding RNA JPX (lncRNA JPX) in the biology of esophageal squamous cell carcinoma (ESCC). The Authors showed that this oncogenic regulator is upregulated in EC9706, Eca109, and KYSE150 compared to normal esophageal epithelial cell line, Het-1A ESCC. The JPX expression is evident in the cytoplasm. The functional assays included in the experimental design showed that JPX knockdown decreases ESCC:
(1) proliferation reducing the number of activly dividing cells in the S phase of the cell cycle and causing their shift toward G0/G1 phase (assays performed on EC9706 and KYSE150);
(2) mobility and invasion what was associated with lowered protein levels of fibronectin, N-cadherin, and vimentin simultaneously with increased levels of E-cadherin
In turn, the upregulation of JPX promoted esophageal tumor growth in vivo and tumor migratory potential.
Furthermore, Authors showed that the potential target for lncRNA JPX is miR-516b-5p. This small non-coding RNA inhibits JPX promoted proliferation and reversed JPX increasing cell migration. Finally, the Authors explained the mechanism of JPX action, showing that this lncRNA promotes esophageal cancer cell growth and mobility via miR-516b-5p/VEGFA signalling pathway.
The article is well-performed, with an excellent experimental design that gives a proof of concept. Indtoduction provides insight into the topic. Methods are adequately described.
The data description and presentation are good. However, the presentation of the data must be improved:
1) Please provide original blots with adjusted molecular weight markers
Answer: We have provide original blots with adjusted molecular weight markers in the supplementary materials.
2) Please provide information regarding the molecular weight of detected proteins
Answer: We have added the molecular weight of detected proteins in figure 3I, 3J and figure 6O.
3) Figure 1D: the quality of microphotographs is poor, especially EdU staining; the scale bar is bearly seen.
Answer: We have changed the microphotographs results in the figure 1D and figure 2D, 2F.
4) Figure 3, 5 and 6: please provide the scale bar for the microscopic images.
Answer: We have added the scale bar for the microscopic images in the figure 3, 5 and 6.
Reviewer 3 Report
The authors present a manuscript regarding the JPX/miR-516b-5p/VEGFA axis in ESCC progression. This regulatory axis in new in ESCC, the methodology is adequate, results and discussion are prepared in comprehensive way, and the results strongly support the conclusions. The manuscript is quite well written, however, it contains several grammar mistakes and typos, therefore it should be revised by an English native speaker, before resubmission.
Other important issues that need to be addressed before manuscript resubmission:
- the manuscript lacks "simple summary" section, which is present in all papers published in Cancers. This is a very short summary (100 - 150 words) that precedes the abstract.
- the (1) reference (global cancer statistics) should be updated to 2022
- The SEM is a measure of precision for an estimated population mean. SD is a measure of data variability around mean of a sample of population. Unlike SD, SEM is not a descriptive statistics and should not be used in this paper. The Authors should recalculate the statistics and express data as mean +/- SD throughout the manuscript
- figures 2D and 2F are barely visible, the Authors should improve microscopic images quality
- figure 2L-O shows the in vivo experiments only with overexpressed JPX. The "opposite" experiment showing the effect of JPX level diminution on tumor volume and weight should be also conducted
-
Figure 3, 5 and 6: scale bars are missing in microscopic images, and in figures legend
- line 305-306: " Furthermore, cotransfection of JPX and miR-516b-5p abolished JPX 305 promoting cell proliferation from 24h to 96 h in EC9706 cells" - I don't understand this sentence. Did the Authors mean the increase of doubling time?
- please unify the PCR abbreviation (qRT-PCR vs RT-qPCR vs Q-PCR)
- densitometric calculations for fig 6O (the intensity of WB bands) would highlight the JPX/mir-516b-5p effects on VEGFA levels
Author Response
The authors present a manuscript regarding the JPX/miR-516b-5p/VEGFA axis in ESCC progression. This regulatory axis in new in ESCC, the methodology is adequate, results and discussion are prepared in comprehensive way, and the results strongly support the conclusions. The manuscript is quite well written, however, it contains several grammar mistakes and typos, therefore it should be revised by an English native speaker, before resubmission.
Answer: The grammar and language have been revised in the manuscript.
Other important issues that need to be addressed before manuscript resubmission:
the manuscript lacks "simple summary" section, which is present in all papers published in Cancers. This is a very short summary (100 - 150 words) that precedes the abstract.
Answer: We have added the simple summary in the manuscript (Line 20-26).
the (1) reference (global cancer statistics) should be updated to 2022
Answer: The (1) reference has been updated (Line 523-524).
The SEM is a measure of precision for an estimated population mean. SD is a measure of data variability around mean of a sample of population. Unlike SD, SEM is not a descriptive statistics and should not be used in this paper. The Authors should recalculate the statistics and express data as mean +/- SD throughout the manuscript
Answer: We have revised the Figures 1-6 and descriptions as mean ± SD in the manuscript.
figures 2D and 2F are barely visible, the Authors should improve microscopic images quality
Answer: Figures 2D and 2F have been changed in Figure 2.
figure 2L-O shows the in vivo experiments only with overexpressed JPX. The "opposite" experiment showing the effect of JPX level diminution on tumor volume and weight should be also conducted
Answer: JPX was found to be highly expressed in esophageal tissues from ESCC patients compared with the adjacent normal tissues. Overexpression of JPX significantly increased tumor volume and tumor weight in a xenograft model. The in vitro and in vivo experiments showed that JPX promoted ESCC gowth, proliferation, and angiogenesis. The results could confirm JPX act as an important oncogenic role in accelerating ESCC progression.
Figure 3, 5 and 6: scale bars are missing in microscopic images, and in figures legend
Answer: We have added the scale bar for the microscopic images in the figure 3, 5 and 6 and figures legend.
line 305-306: " Furthermore, cotransfection of JPX and miR-516b-5p abolished JPX 305 promoting cell proliferation from 24h to 96 h in EC9706 cells" - I don't understand this sentence. Did the Authors mean the increase of doubling time?
Answer: We have revised this sentence in the results (Line 364-365). It indicates transfection time.
please unify the PCR abbreviation (qRT-PCR vs RT-qPCR vs Q-PCR)
Answer: We have unified the PCR abbreviation in the manuscript (Line 213, line 715, line 721).
densitometric calculations for fig 6O (the intensity of WB bands) would highlight the JPX/mir-516b-5p effects on VEGFA levels
Answer: We have added the band intensity in figure 6O.
Round 2
Reviewer 1 Report
Inseted examinations (Figure 2P, 2Q) and reference (20) are adequate and helpful for readers.